# The impact origin and evolution of Chryse Planitia on Mars revealed by buried craters

Lu Pan [1], Cathy Quantin-Nataf[1], Sylvain Breton[1] & Chloé Michaut [1]

Large impacts are one of the most important processes shaping a planet's surface. On Mars, the early formation of the Martian crust and the lack of large impact basins (only four unambiguously identified: Hellas, Argyre, Utopia, and Isidis) indicates that a large part of early records of Mars' impact history is missing. Here we show, in Chryse Planitia, the scarcity of buried impact craters in a near-circular area could be explained by a pre-existing topographic depression with more intense resurfacing. Spatially correlated with positive Bouguer anomaly, this near-circular region with a diameter of ~1090 km likely originated from an impact. This proposed large impact basin must have been quickly relaxed or buried after its formation more than 4.0 billion years ago and heavily modified by subsequent resurfacing events. We anticipate our study to open a new window to unravelling the buried records of early Martian bombardment record.

---

[1] Université de Lyon, Université Claude Bernard Lyon 1, ENS de Lyon, CNRS, UMR 5276 Laboratoire de Géologie de Lyon -Terre, Planètes, Environnement, 69622 Villeurbanne, France. Correspondence and requests for materials should be addressed to L.P. (email: lu.pan@univ-lyon1.fr)

Large basin-forming impacts significantly modified planetary surfaces and crusts, especially in the first billion years of solar system history when the impactor flux was orders of magnitude higher than at present[1]. On Mars, a hemispheric dichotomy formed during this period, possibly as a result of one giant impact[2,3], resulting in a dramatic elevation difference of 5–6 kilometers between the southern highlands and northern lowlands. Large basin-forming impacts have played a major role in shaping Mars' surface and its climate, through interactions with the hydrosphere or cryosphere[4,5]. However, the early cratering record on Mars is not as clear as the Moon and Mercury since it has been modified via erosion and resurfacing events. Although the Martian crust may have formed within a hundred million years from solar system formation[6,7] during which impact flux was expected to be high, currently on Mars only four large basins (diameter >780 km) have been identified with confidence (Hellas, Argyre, Utopia, and Isidis) among which only Isidis impacted onto the dichotomy. The observed cratering record may imply a decrease in the impactor flux during Mars' post-accretionary phase[8]. Alternatively, many large basins could have existed on Mars[9–11] but the evidence that commonly supports the identification of large impact basins are obscured by later resurfacing, lending some uncertainties to their identification. The confirmation of these ancient basins would advance our understanding of the impactor flux in the early period before ~4 Ga and place crucial constraints on the timing of the dichotomy forming event and the thermal evolution of Mars.

Impact basins have near-circular topographic depressions with surrounding ringed structures, and mostly show positive Bouguer anomalies in gravity data. Extensive studies on the moon showed circular, positive Bouguer anomalies form in impact basins as a result of a combination of crustal thinning, uplift of the mantle beneath, and later infill of dense materials[12–15]. These features in early basin-forming impacts may be obscured by subsequent crustal relaxation and burial beneath Hesperian and Amazonian aged lava-plains and sediments, especially within the northern lowlands of Mars[16,17]. Chryse Planitia is a ~2000 km wide basin within the northern lowlands near the dichotomy, located at the termini of Mars' most prominent outflow channels (e.g., Ares Valles and Kasei Valles). An impact origin for the basin has been proposed previously[9,18,19], but was debated since there is no clear topographic depression compared to the adjacent lowlands, the gravity signature appeared weak and non-circular and any surface feature has been removed. With the recent advent of high-resolution gravity models[20] and comprehensive imagery[21] and topography data[22] of the planet, we revisit the possibility of Chryse Planitia being originated as an impact basin. Instead of the surficial geologic features, we investigate the subsurface pre-fill topography in Chryse Planitia using the morphology and distribution of buried craters.

Impact craters have predictable initial morphologies, depending on target properties and impactor mass and velocity[15]. Crater depth and rim height, subject to erosion and infill, decrease as varying functions of time, and the number density of a crater population can be predicted by the time-integrated impactor flux assuming a production and chronology function[23–25]. The observed morphology and distribution of impact craters at present, as a consequence, can be used to trace the extent, timing and intensity of crater modification processes in the past[26–28], as well as to constrain the thicknesses of specific geologic deposit[29–32]. Here we identify distinct crater populations within a circular area in Chryse compared to the adjacent region, which could be related to a more intense resurfacing event in a topographic depression. This region also shows a quasi-circular positive gravity anomaly, which combined with the buried topography from impact crater statistics, suggests that the circular region could be the original Chryse impact basin, which subsequently underwent heavy degradation.

## Results

**Impact crater populations of Chryse and Acidalia region.** Several categories of crater morphology that differ from pristine impact craters have been identified on Mars since the era of Mariner 9 images, including flat-floor craters and even more degraded ghost craters[33]. With higher resolution images, a greater variety of modified impact crater morphology has been resolved. Although multiple classes of degraded craters are found in the southern highlands[34], most of the impact craters in the northern lowlands can be classified into two distinct groups, the relatively pristine craters, and the degraded craters. The degraded craters have been previously documented based on different criteria of identification: e.g., quasi-circular depressions (QCD), sometimes with corresponding positive Bouguer anomalies (previously recognized as crustal thinned areas (CTA))[11,18,35–37]; and smaller, filled stealth craters widespread in the extent of the Vastitas Borealis Formation[17]. While these results clearly suggest different levels of burial in the northern lowlands, the identification of these degraded impact craters may be challenging since some local depressions from sublimation, collapse or volcanic eruption could be misidentified as impacts due to the high degree of degradation and lack of geologic context. To reconcile the different criteria for identification of buried impact craters, we extract previously mapped impact craters within the Chryse and Acidalia region[38,39] and systematically classify them based on their confidence of identification and degrees of burial (Methods, Crater morphology classification).

Although the surface topography has no sign of a circular basin in the current Chryse/Acidalia region, the Goddard Mars Model 3 Bouguer gravity map[20] shows two near-circular positive Bouguer anomalies in the southwest part of Chryse Planitia (Fig. 1) that are denoted as Chryse anomaly unit and Small anomaly unit thereafter. Initial mapping of buried impact craters shows their distribution is highly heterogeneous within Chryse and Acidalia Planitia regardless of their degrees of degradation (Fig. 1). This heterogeneous distribution does not correlate well with geological units mapped previously based on the morphology of surficial units[40] and likely reflects a different modification history of the subsurface. Following the gravity contours, we extract the depth and diameter information for all the impacts in the two anomaly units as well as in two adjacent units outside the prominent positive Bouguer anomaly (Fig. 2). The bimodal distribution of the depth-diameter ratio in all of these regions confirms the existence of two distinct crater populations (Fig. 2b, c), one relatively pristine population following the expected depth-diameter relationship and the other representing buried impact craters with much-reduced depth compared to pristine craters. Regions correlating with a positive Bouguer anomaly (Chryse and Small anomaly units) have a smaller probability density of buried craters relative to pristine craters, whereas in the adjacent regions, the probability density of buried craters is much higher than that of the pristine craters. This discrepancy in crater populations in these different units could be related to distinct resurfacing histories in these two types of regions.

**Crater degradation and modification.** Since the fresh crater population in the anomaly and adjacent units follows a common trend in the depth-diameter relationship (Fig. 2b, c), there is likely negligible difference in the initial morphology of impact craters. Continuous, non-variant obliteration in the history of Mars is inconsistent with the two distinct types of crater populations (Fig. 2c, d) observed within both types of units, since a constant

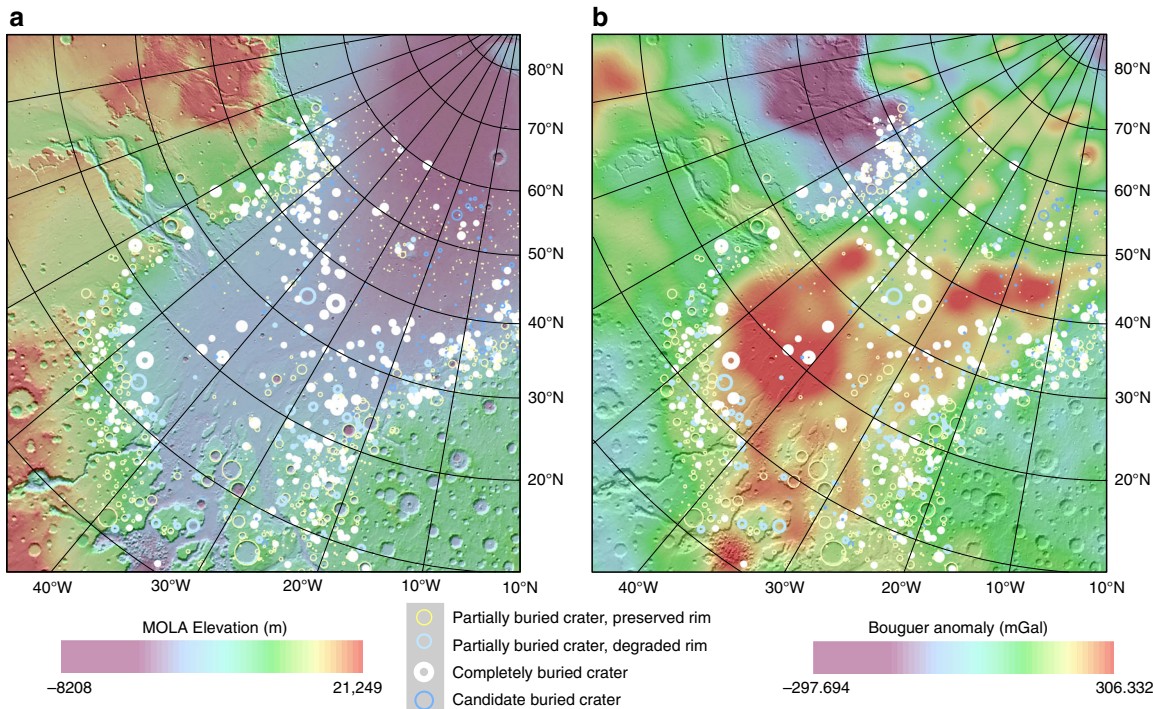

**Fig. 1** Map of buried crater distribution within the Chryse and Acidalia Planitia overlain on **a** Topography from Mars Orbiter Laser Altimeter (MOLA) data;[22] and **b** Goddard Mars Model 3 gravity data, Bouguer anomaly[20]. The base maps are projected using the Mars North Polar Stereographic Projection

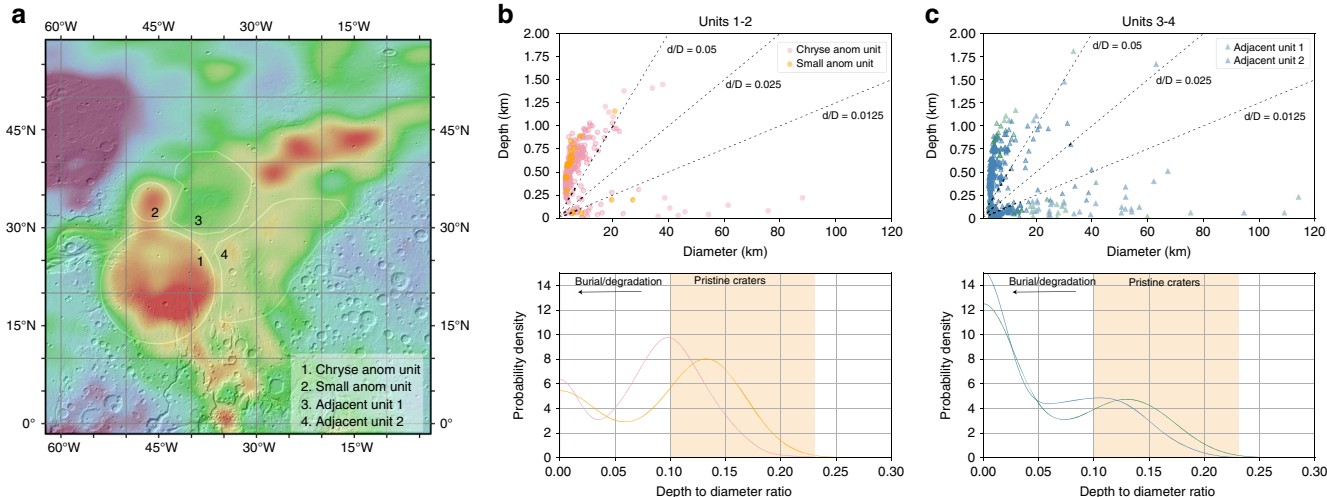

**Fig. 2** Depth-to-diameter ratio distribution in different units within the Chryse Planitia. **a** Bouguer gravity anomaly map[20] of the Chryse and Acidalia Planitia showing the location of the anomaly units and adjacent units in Mars Simple Cylindrical Projection. **b, c** The depth-diameter scatter plot (upper) and probability density distribution of depth/diameter ratio (lower). **b** the Chryse anomaly unit and Small anomaly unit; **c** The Adjacent units 1 and 2. Shaded regions indicate the distribution of the depth-to-diameter ratio of pristine craters

obliteration rate only permits a unimodal impact crater population (Methods, Crater accumulation and modification). The observed bimodal populations indicate a catastrophic resurfacing event early in Mars' history where many impact craters are modified by infill and/or erosion, resulting in reduced depths. The different probability densities of craters of given depth-to-diameter ratio observed within the anomaly units compared to adjacent units may be due to a difference in the timing or intensity (i.e., differential infilling or erosion) of the catastrophic event.

Comparing Chryse anomaly unit and Adjacent unit 1, the size-frequency diagram of all impacts (pristine and degraded)

shows sudden changes in slope or kinks (Fig. 3a), which are indicative of resurfacing events or changes in obliteration rate[23] assuming a constant crater production function over time. With comparable overall crater areal density, the anomaly unit has fewer large impacts but more small impacts than the adjacent units. If the crater production remains unchanged, the kink at around 30 km could be interpreted as a major resurfacing event around 3.7 Ga. For craters smaller than 7 km the size-frequency distribution continually cross isochrons, probably due to continuous crater obliteration process[41] or a secondary, small scale resurfacing event until ~3.4–3.5 Ga for both units.

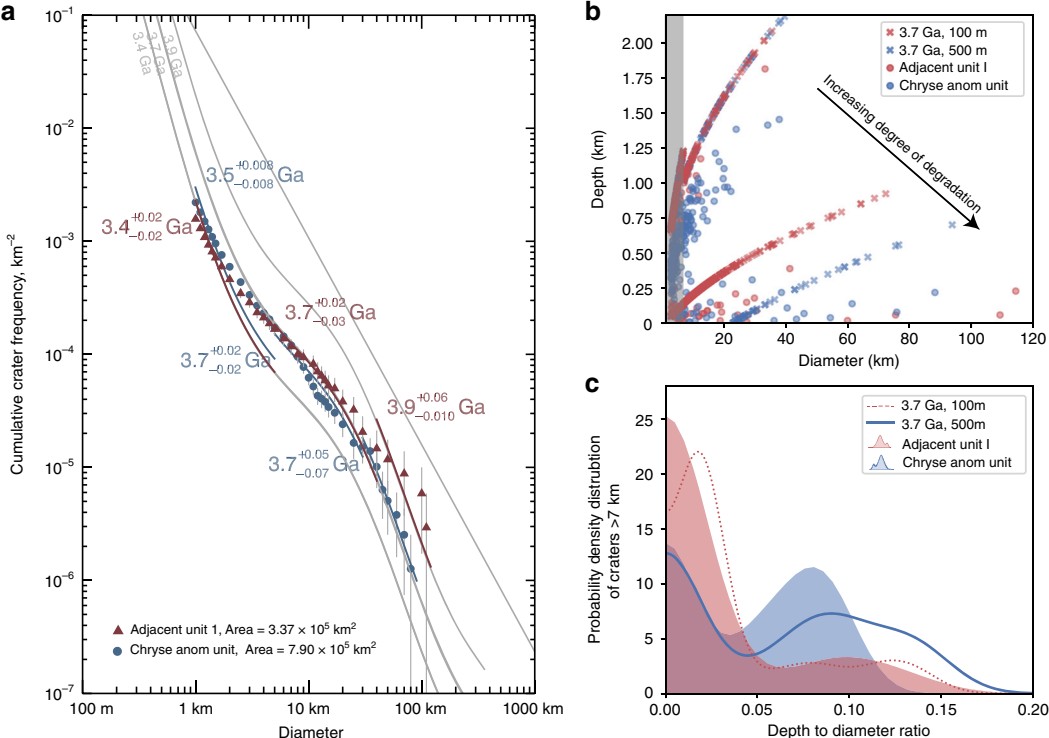

**Fig. 3** Impact crater size frequency and crater depth to diameter ratio compared to crater accumulation models. **a** Cumulative fit on the crater size-frequency diagram showing the periods of major resurfacing events in the Chryse anomaly unit (blue) and the Adjacent unit 1 (red) using Craterstats[23]. The isochrons of 3.4, 3.7, and 3.9 Ga are plotted in grey lines, calculated based on the crater production function[24] and chronology function[25]. The tabulated data are attached in the Supplementary Tables 2–3. **b, c** Modeled impact crater morphology and probability density for a common instantaneous resurfacing event around 3.7 Ga with zero obliteration rate, compared to observations. **b** The depth-diameter relationship of modeled results (crosses) compared to observations (circles) with craters smaller than 7 km masked in the shaded zone. **c** The probability density distribution of depth to diameter ratio as a proxy for the relative abundance of modeled pristine and filled impact craters (non-shaded) compared to observations (shaded). The distribution is calculated for craters larger than 7 km

To investigate the evolution of crater morphology and crater population with time, we model the crater population by incrementing, at each timestep, impact craters with an initial morphometry predicted by the morphometry statistics of Martian impact craters[42], assuming a known impactor flux as a function of time[24,25] (Methods, Crater accumulation and modification), starting from 3.9 Ga. Since the preserved rim and flat-floored morphology of most buried craters in the region clearly suggests that burial by either volcanic or sedimentary deposits is a dominant process[17,35,36] (Supplementary Note 1, Supplementary Fig. 1), we consider a resurfacing event of varying thicknesses occurring at 3.7 Ga that leads to a reduction of the rim height and crater depth and to the removal of small impact craters that are completely filled (i.e., their rim heights are smaller than the thickness of fill). The crater population and morphometry are also modified by a long-term obliteration process, with reduced crater depth at each timestep. The model results are compared to the observation for craters larger than 7 km assuming the smaller impact craters are mostly affected by secondary modification or infill. Figure 3b, c shows the resulting crater population when there is no obliteration, with a differential fill at 3.7 Ga with 100 m and 500 m thickness. In the case of zero obliteration rate, the observed population of buried craters in Chryse anomaly unit can be best fit by a one-time event with fill thickness larger than 500 m, whereas Adjacent unit may require a smaller fill thickness or the modification of a secondary resurfacing event. Although the obliteration rate and secondary infilling events may be important for smaller impact craters (Supplementary Note 2; Supplementary Fig. 2), the difference in resurfacing history between the Chryse

and adjacent units plays a primary role in shaping the crater morphometry distribution for larger impact craters. The infilling of fluvial or volcanic deposits within the Chryse anomaly unit is consistent with a pre-existing topographic depression in the circular anomaly unit centered at 315°E 21°N, with a diameter of ~1090 km.

**Estimate of the thickness of fill.** In order to understand the evolution of the possible Chryse basin, we estimate the average thickness of fill directly based on the morphometry of buried impact craters, either partially or completely buried, as a function of distance to the center of the basin. We treat completely buried impact craters similarly as partially buried craters (Fig. 4a, b) (See Supplementary Note 3), assuming the excess of fill (Δ) is small, allowing visual observation on MOLA topography.

The expected fill is then predicted as shown in Eq. (1).

$$T_{fill} = R_h(D) - d \qquad (1)$$

here $T_{fill}$ is the thickness of fill; $d$ is the observed crater rim-to-floor depth; $D$ is the observed diameter of the crater; $R_h$ is the original rim height calculated from the statistics of fresh crater rims in Acidalia Planitia[42].

The average thickness of fill for partially filled craters does not vary as much as that for the completely filled craters (Fig. 4). Unlike partially filled craters, the completely filled craters are less prone to aeolian and fluvial erosion. In particular, the estimated thickness of fill for completely filled craters increases immediately within the radius of the anomaly unit, with a slight decrease

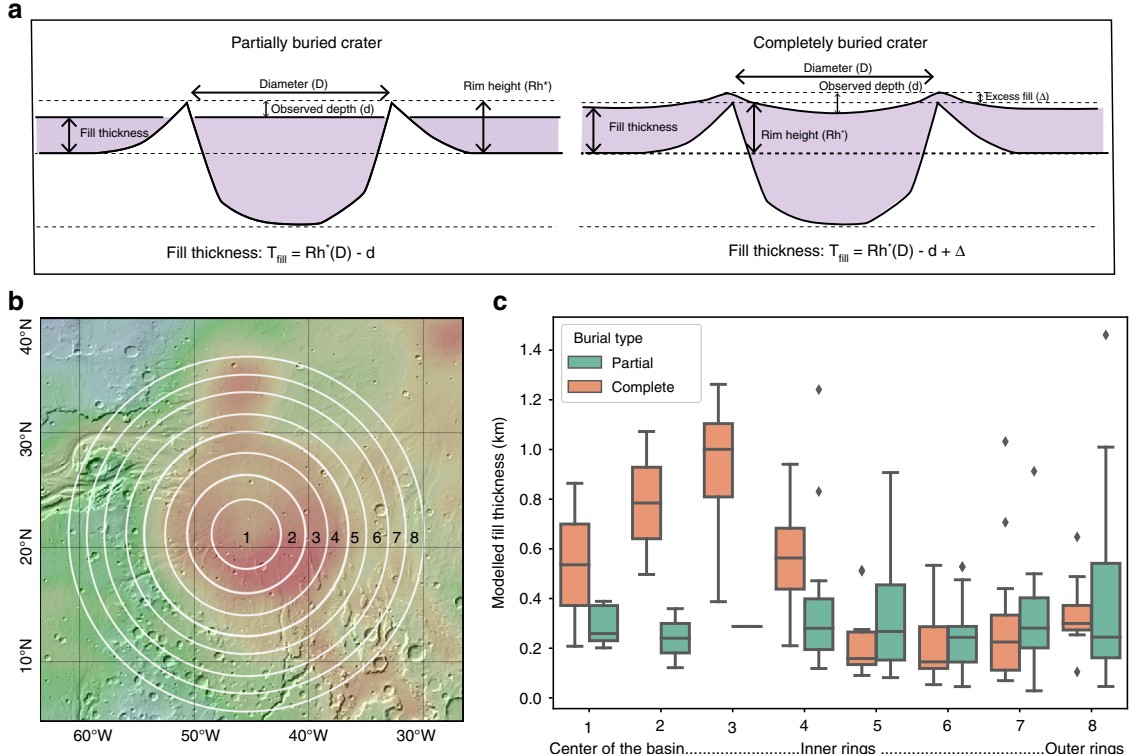

**Fig. 4** The estimated thickness of fill based on crater morphometry in the region of Chryse Anomaly unit. **a** Schematics of the morphological parameters used to calculate fill thickness for partially and completely buried craters. **b** Geographic locations of the regions for the calculation of impact crater thickness in **c**, with Mars Bouguer gravity anomaly as the base map[20]. **c** Variation of the calculated fill thickness with distance to the center of the anomaly. The colored box indicates the range of values from the lower to upper quartile for each column, with a black horizontal line at the median value. The whiskers extending from the box show the total range of the data, except for outlier points which are indicated using black diamonds

toward the center of the unit. This increase inside the anomaly unit is indicative of an increase of fill up to ~600–1000 m, similar to our previous estimate, whereas outside the anomaly the fill thickness ranges from 150–350 m.

## Discussion

Our two different methods point to an infilling of more than 500 m within a rather flat circular area that coincides with a quasi-circular-positive Bouguer anomaly. This suggests the existence of a pre-fill topographic depression. The correlating gravity anomaly is heterogeneous within the unit with a maximum of ~350 mgal. Assuming a simple plateau model, such a large amplitude (Supplementary Note 4, Supplementary Fig. 3) cannot be produced by an infill of up to 1 km of dense basaltic lava flows, and thus requires a thinned crust probably due to mantle uplift in a large impact. We consider that an impact is a possible origin for the anomaly unit within Chryse Planitia. In analogy to other impact basins on the Moon, the observed circular region, which has a diameter ($D$) of ~1090 km, would represent the diameter of the inner ring of the potential impact basin; this would indicate that the main rim has a diameter of at least $\sqrt{2}D$ (1500 km)[15,43].

Interestingly, both the amplitude of topographic depression and the gravity signature are small compared to the expected values for an impact basin of this size. For example, Isidis basin (~1500 km in diameter) also formed at the Martian dichotomy, has a moderate amount of degradation or fill. Isidis has a depth around 4–5 km, and a gravity anomaly maximum of ~850 mgal, in contrast to the depth of 0–1 km and a gravity anomaly of ~350 mgal for Chryse. The gravity anomaly in Chryse is much smaller than Isidis and Argyre but close to that of Utopia[44], another

buried impact basin within the northern lowlands. The diminished gravity and small topographic expression (observed from impact craters populations) both indicate that the basin was quickly relaxed and filled after the impact. Since Chryse basin probably formed early in Mars' history, the impact structure could be prone to viscous relaxation, due to higher geothermal gradient[45,46]. Alternatively, the basin could have been quickly filled up since the impact with low-density sedimentary deposits, possibly originating from the early activity of the outflow channels[16]. Filling can also attenuate the gravitational signature from the uplifted mantle. Assuming an original impact basin with 5–8 km in depth, the volume of low-density sedimentary deposits would be $1.4–2.3 \times 10^7$ km³, an order of magnitude higher than the total estimated volume of eroded sediments in the circum-Chryse outflow region[16,47,48]. It is thus unlikely that infill of sedimentary deposits is the sole reason for the reduced gravity anomaly within the basin, whereas this reduced gravity anomaly signature could have been a common feature of early impact basins on Mars like Utopia and Chryse.

The oldest age (~4.0 Ga) within the Chryse basin units is found in a small region near the dichotomy[49,50], which sets the lower limit of the age of the basin. Considering both the degradation state and the inferred degree of relaxation, it is likely that Chryse is older than the better-preserved large basins on Mars (Isidis, Argyre, and Hellas). Utopia basin, on the other hand, has diminished gravity signature and topographic expression similar to Chryse, which may have formed during the same period. Chryse impact overlies the dichotomy boundary, so the age of the impact basin also constrains the formation age of the dichotomy and indicates that the dichotomy formation event predates probably the oldest surfaces on Mars, as suggested previously[18,19,50]. Chryse

basin-formation event (and other buried impact basins on Mars) is probably representative of the time period between the dichotomy formation and the rest of Mars' surface.

Early energetic impacts could have heavily fractured the upper crust of Mars[51], remobilized groundwater at the time of the impact, which would initialize catastrophic floods and outflow channels or induce regional hydrothermal systems[4,5]. Identification of impact basins that may have been relaxed and/or buried like Chryse is crucial to understanding the unknown early impactor flux before the period of 3.8–4.1 Ga, and would also provide further insight on how these impacts have modified Mars' crust and influenced the early climate. Other than Chryse anomaly, the small circular anomaly unit to the north of Chryse is likely another previously unknown impact basin with a diameter of 500 km. Other possible large basins, such as Acidalia[9], Daedalia[52], could also have formed in the early era of Mars and their surficial expressions largely modified in a similar way by later erosion or volcanic infill. Lunar impact basins show a change in relaxation state between 4.2–4.4 Gyr, which has been interpreted as the cooling of the base of the lunar crust[53–55]. Since a wet rheology and a hotter interior both favor crustal relaxation, the same type of change in the relaxation state may have occurred later on Mars. A thorough investigation into the ages and relaxation state of these large impact basins on Mars may thus reveal important constraints on the thermal evolution of the planet. We expect the buried records of the earliest epoch of Mars' history would be revealed with further detailed analysis involving depth as the third dimension in crater statistics, facilitating our understanding of this important period of Mars' history.

## Methods

**Crater morphology classification.** To map the impact craters in the Chryse and Acidalia Planitia, we first subtracted all the impact craters within the region of the northern lowlands[48] from the Mars crater database[38,39]. A small set of circular features likely with an impact origin that were not included in the database are also examined. These craters are then classified into five categories according to their observed morphology: partially filled craters with a pristine rim structure; partially filled crater with a degraded rim, completely filled craters, candidate filled craters, and unclear circular features.

The workflow is as follows:

(1) First, we identify if the impact craters have been substantially filled. We use both a cutoff threshold for a depth-to-diameter ratio of 0.05 as well as visually checking their morphology on the THEMIS global mosaic[21]. Only impact craters with substantial fill that significantly differ from the initial morphology of fresh impact craters are included.

(2) These degraded impacts are further classified into partially and completely buried depending on the exposure of crater rim. If more than 50% of the crater rim is visible and preserved, they are classified as partially buried craters with preserved rim. If less than 50% but more than 10% of the crater rim is preserved, we classify them as partially filled craters, with degraded rim.

(3) For those circular features with less than 10% rim visible, we investigate their topographic expressions using MOLA Mission Experiment Gridded Data Records (MEGDR) (for craters larger than 5 km) and MOLA Precision Experiment Data Record (PEDR) data profile (for craters smaller than 5 km). If more than 50% of the raised rim is observed in the topographic data, the crater is classified into completely buried crater.

(4) A near-circular depression feature with less than 10% of rim exposed in imagery and less than 50% of raised rim visible in topographic data would be classified as candidate buried crater.

(5) If a topographic depression is observed and mapped in the crater database[38,39], but the crater rim is not obvious and the topographic expression is non-circular, we classify them as unclear category.

We show a type of example for each category (Supplementary Fig. 1). Although these classifications can be well diagnosed within the Chryse region and its surroundings, this classification regime does not account for all crater morphologies outside of the study region and is not intended to distinguish different degradation processes. For example, many degraded impact craters in the southern highlands[34] show both a retreating, rounded crater rim and flat crater floor, consistent with fluvial or wind erosion rather than volcanic infill. Within the Vastitas Borealis Formation (VBF), smaller infilled craters that stand above the surrounding terrain often have preserved crater ejecta. These infilled craters previously referred to as pedestal craters[56,57], show distinct morphology to the partially buried impact craters in the Chryse region. Although these are classified as

partially degraded impact craters, they have definitive morphological features indicating degradation processes likely due to sublimation of ice-rich subsurface terrain, which is not related to the degradation process within Chryse Planitia.

**Kernel density estimation.** We estimate the crater probability distribution with kernel density estimators[58–60] instead of binned frequency distributions, to account for the underlying uncertainties of the impactor population, impactor to crater scaling and degradation process, including, but not limited to: varying impactor and target conditions; variation of infilling/erosion processes spatially; and the errors on depth and diameter measurements. This distribution estimated by kernel density in theory shows the same shape as the binned frequency distributions, with a similar approach previously suggested to better present and analyze crater size-frequency distribution[61]. This also enables assessment of a continuous distribution with no prior assumptions about the distribution of the sample. Here the estimated probability distribution $\widehat{f_h}(x)$ is given as:

$$\widehat{f_h}(x) = \frac{1}{Nh} \sum_{i=1}^{N} K\left(\frac{x - x_i}{h}\right) \quad (2)$$

where N is the total number of samples, h is the bandwidth and K is the kernel function chosen. The type of kernel chosen for this estimation does not change the overall shape of the density distribution[60,61]. Here we choose a Gaussian function as kernel, so we have:

$$\widehat{f_h}(x) = \frac{1}{Nh} \sum_{i=1}^{N} \frac{1}{\sqrt{2\pi}} e^{-\frac{(x - x_i)^2}{2h^2}} \quad (3)$$

The effect of varying bandwidth changes the range over which the probability density distribution function would be smoothed. Since the errors in the measurements of impact crater depth on different Martian datasets have not been addressed in detail, we here test a variety of bandwidths including the non-parametric rules of thumb method[60] and taking constant bandwidths (e.g., h = 0.1–0.5). Smaller bandwidths create spikes in the distribution and thus increase the variance, since the kernel functions do not overlap with adjacent points. As a general case, we here apply the rules of thumb normal method[60], where the bandwidth h is estimated by:

$$h = 1.059 * \sigma * n^{-\frac{1}{5}} \quad (4)$$

σ is the standard deviation of the sample.

An intrinsic artifact to apply kernel density estimation to depth-to-diameter ratio (d/D) is that the function (Eq. 2) predicts positive probability for negative values near d/D = 0. Since we don't expect the depth of a crater to be negative, the probability distribution of d/D should be discontinuous beyond x = 0, which creates a boundary at x = 0 for the probability density function. To account for this boundary effect, we calculate the ordinary kernel estimate based on augmented data[60] ($-x_n, \ldots, -x_2, -x_1, x_1, x_2, \ldots, x_n$), with a bandwidth based on the original sample size N, so that the probability of negative values is accounted for near x = 0. Such modification corrects for the probability of negative depth to diameter ratios and thus presents a better estimate to the probability density function of depth-to-diameter ratio of impact craters.

**Crater accumulation and modification.** We implement a numerical model to account for the impact crater production and modification processes. We first generate synthetic crater size and depth distributions assuming a fresh surface at the starting time $t_0$ = 3.9 Ga, set by the oldest surface age based on large impact craters (D > 30 km) in the size-frequency diagram (Fig. 3). We investigate here a crater diameter range from 3 to 1000 km, where the smallest crater size corresponds to the crater diameter in the observation limited by MOLA data resolution and the largest is set by the area investigated. The diameter range is distributed into 50 bins in the logarithm space. The number of craters added for each diameter bin D at age t for a unit with area A follow a Poisson distribution with an average number $\bar{n}$:

$$\bar{n}(D, t) = A * N(D, t) \quad (5)$$

Here N(D, t) is the crater number density of a given diameter bin and time, which can be separated into two functions: the production function f(D), characterizing the number density of given diameter bin, and $N_1(t)$ the number of craters larger than 1 km at a given time $t$[25].

$$N(D, t) = f(D) * N_1(t) \quad (6)$$

The number density of given diameter can be calculated following the polynomial:

$$log_{10}f(D) = a_0 + \sum_{n=1}^{11} a_n [log_{10}(D)]^n \quad (7)$$

Here $a_n$ is one of the 12 coefficients for the polynomial (Supplementary Table 1). $N_1(t)$ is set by the chronology function:[25]

$$N_1(t) = 2.68 \times 10^{-14} (e^{6.93t} - 1) + 4.13 \times 10^{-4} t \quad (8)$$

We model the number density of craters in its time derivative form:

$$\partial N(D,t)/\partial t = f(D) * N_1'(t) \tag{9}$$

Thus, with the initial condition $N(D, t_0) = 0$, for each timestep $\Delta t = 0.01$ Ga, we model the incremented number of impact craters and integrate between $t$ and $t+\Delta t$[24].

$$\Delta N(D,t) = f(D)[N_1(t + \Delta t) - N_1(t)] \tag{10}$$

Given the number of incrementing impacts at each timestep for each diameter bin, we compute their initial crater depth and rim height following geometric relations to their diameter for simple and complex craters, respectively. We adopt a transition diameter of 7 km from simple to complex craters for Mars from Mars global crater statistics[39,62]. Within Chryse Planitia, the surface geologic units do not indicate distinct materials in the anomaly and adjacent units, so the same geometric relationship should apply. However, significant variabilities in crater morphometry have been observed in previous work between the northern and southern hemispheres, which has been attributed to different target properties[39,42]. To account for this difference, we apply the impact crater morphometry based on fresh craters in Acidalia Planitia, adjacent to Chryse[42]. The depth (d) is scaled with the diameter (D) for simple craters as follows:

$$d(D) = 0.302D^{0.72} \dots (D < 7km) \tag{11}$$

d(D) for complex craters ($D \geq 7\,km$) were missing in the original reference (personal communication) so we calculate the rim-to-floor depth by adding the surface-to-floor depth ($d_s$) and rim height ($H_R$) of the crater.

$$d(D) = d_s(D_S) + H_R(D) = 0.384D_S^{0.38} + 0.072D^{0.62} \dots (D \geq 7km) \tag{12}$$

The diameter at the pre-impact surface $D_s$ is scaled to the rim-to-rim diameter

$$\frac{D_R}{D_S} = 1.09D_R^{0.013} \tag{13}$$

Therefore, we have:

$$d(D) = 0.372D^{0.375} + 0.072D^{0.62} \dots (D \geq 7\text{km}) \tag{14}$$

The initial rim height, relevant in the resurfacing process, is calculated as followed[42]:

$$R_h(D) = 0.082D^{0.54} \dots .(D < 7km) \tag{15}$$

$$R_h(D) = 0.072D^{0.62} \dots .(D \geq 7km) \tag{16}$$

Taking different assumptions on crater initial morphometry relations results in only minimal variations on the resulting crater population (Supplementary Note 5, Supplementary Fig. 4). The depth-to-diameter ratio following the above relationship is 0.17–0.3 for simple craters and 0.04–0.13 for complex craters smaller than 50 km. Assuming a distribution of crater diameter as predicted by the production function[24], the depth-to-diameter ratio follows a distribution dominated by small craters with a mean depth-to-diameter ratio at 0.19–0.2. Fresh complex craters are clustered at 0.1, but they are of smaller probability densities compared to the simple craters. With constant erosion and infill that affects small craters more easily, the depth-to-diameter distribution of a single geologic unit evolves into a left-skewed, unimodal distribution. The depth and rim height for craters between 7–10 km may be underestimated since impact craters in this transitional diameter range on Mars may fall into both simple and complex craters due to local varying target property.

We model the crater modification process with a constant obliteration rate (b) such that the impact crater depth ($d_t$) is reduced at each timestep by the product of obliteration rate and time interval ($\Delta t$). The obliteration rate is in the unit of m/Gyr.

$$d(t) = d_0 - \int_{t_0}^{t} b dt \tag{17}$$

In the case for a one-time infill event, we assume the upper surface of the infill of crater depth to be at equal elevation with the infill surrounding the impact craters (fill thickness) (Fig. 4a). At a specific timestep (e.g., 3.7 Ga), a one-time infill event would reduce the rim height ($R_h$) and crater depth (d) accordingly:

$$R_h' = d' = R_h(D) - T_{fill} \tag{18}$$

$R_h'$ and $d'$ represent the new rim height and depth after infill; $t$ is the thickness of the unit.

After each timestep, impact craters are removed if their depth is equal to or smaller than zero. As such, the time series of crater addition and obliteration continues until the present (0 Ga). The resulting crater size, morphology, and distribution are output to understand the effect on crater population from different obliteration processes. The sensitivity to varying starting time and obliteration rates are tested and further discussed in Supplementary Note 6 (Supplementary Fig. 5). We calculate the chi-square statistic for the number of buried impact craters (d/D < 0.04) with a diameter larger than 7 km in 10 linear diameter bins, so that we could evaluate if the model reproduces the observed crater population by minimizing the chi-squared value.

$$\chi_c^2 = \sum \frac{(N_{model} - N_{data})^2}{N_{data}} \tag{19}$$

where $N_{model}$ and $N_{data}$ are the number of craters in each diameter bin based on the density distribution of the model and the data, respectively.

## Data availability

The Mars data analyzed in this study are publicly available from the Geosciences Node of the Planetary Data System (http://pds-geosciences.wustl.edu/). The analysis that supports the findings of this study is available from the corresponding author upon reasonable request.

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

## Acknowledgements

This research has received funding from European Union's Horizon 2020 research and innovation program under the Marie Sklodowska-Curie Actions grant agreement 751164 to L.P. C.M. acknowledges financial support from the IDEXLyon Project of the University of Lyon in the frame of the Programme Investissements d'Avenir (ANR-16-IDEX-0005). The authors also acknowledge insightful discussions with Suzanne Smrekar.

## Author contributions

L.P. conceived the study, performed the data analysis, and prepared the early draft of the manuscript. C.Q. supervised the project and participated in early discussions. S.B. provided the original model for impact crater accumulation and modification. C.M. assisted in the gravity data interpretation. C.Q. C.M., and S.B. contributed to science discussions and revision of the manuscript.

## Additional information

**Competing interests:** The authors declare no competing interests.

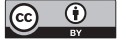

