## [Peer Review File · Nature Communications]

REVIEWERS' COMMENTS:

Reviewer #1 (Remarks to the Author):

Below please find my summary review of NCOMMS-19-15983-T, The impact origin and evolution of Chryse Planitia on Mars revealed by buried craters, by Pan et al.

I previously reviewed this manuscript as a for Nature Geoscience. I trust you have this previous review; my opinion was that this paper is clearly presented and provides an analysis of high quality, but that the results and conclusions do not rise to the level of newsworthiness I believe the journal is looking for because they are not novel.

In the revised paper and response the authors have modified the description of their degradation model and provided two possible implications of the work: (i) They suggest their model has the potential to better quantify erosion and infill, and (ii) They confirm the origin of Chryse Planitia as an early impact basin.

After rereading the paper and considering its implications, I am afraid I am still concerned that these results do not merit publication in Nature Communications. (If anything, I think the work is even less appropriate here than for Nat Geo because this journal addresses a broader audience for whom novelty and broader implications should be even more important.)

The paper argues that recognizing Chryse as an impact basin could cause radical changes in our interpretations of events on early Mars could be used to constrain mantle rheology and thermal evolution based on the relaxation state. But such profound conclusions are not presented here. What the paper does offer is a very nice analysis which strengthens (though not totally proves) a previously suggested interpretation. I am sorry I therefore I cannot recommend publication in Nature Communications. I recommend the authors send their work to one of the main stream, high quality journals in Planetary Science, and I do believe it should be published in such a journal.

Reviewer #2 (Remarks to the Author):

The reviewer is highly pleased with the extensive MS reworking by authors. Now most of reviewer's concerns are answered. The main idea of the MS – the possibility of the impact origin of the Chryse depression and fast filling of the original depression with sediments – is now presented in better style than in the first MS' variant.

However, a couple of minor problems of the paper perception still exist. In reviewer's opinion, the submitted version 2.0 is good enough for publication, but not meet yet standards of the Nature Communication. The recommendation is a minor editing without the next review (see the wishlist below).

1. I would advise to change the word "density" to "probability density" through the whole text (both main and supplementary). A general reader in planetology commonly uses "density" or as a physical density, or an "areal density" (number of craters per unit area). Authors frequently have in mind the statistical probability density (the fraction of craters with a given d/D in the narrow interval of d/D).

2. The usage of nonparametric probability density estimation has the sense when we have underlying ideas what is the reason for the data scatter. It may be a statistical error of measurements, the difference in the crater degradation rate in various places (sedimentation could be faster in local depressions), or whatever else. It would be great if in Supplements authors extend the explanation of their statistical technique choice. Just now it looks like they have a standard (MATLAB?) computer program and use it because they have it. In the previous review I pointed out the presence of $d/D < 0$ in the statistical processing. Now authors simply put all virtual numbers of craters with $d/D < 0$ into the first (?) d/D bin next to $d/D = 0$. Why authors name this "improvement" as a "reflection technique"? Is it possible to show for the comparison a simple statistics of d/D (for example in a style used for lunar craters by Basilevsky, A. T., Kreslavsky, M. A., Karachevtseva, I. P., and Gusakova, E. N., Morphometry of small impact craters in the Lunokhod-1 and Lunokhod-2 study areas, Planet. Space Sci., 2014. V. 92. P. 77-87)? It is non-mandatory, sure, but a short paragraph in Supplementary would greatly improve the paper perception.

Response to referees

Reviewer #1 (Remarks to the Author):

Below please find my summary review of NCOMMS-19-15983-T, The impact origin and evolution of Chryse Planitia on Mars revealed by buried craters, by Pan et al.

I previously reviewed this manuscript as a for Nature Geoscience. I trust you have this previous review; my opinion was that this paper is clearly presented and provides an analysis of high quality, but that the results and conclusions do not rise to the level of newsworthiness I believe the journal is looking for because they are not novel.

In the revised paper and response the authors have modified the description of their degradation model and provided two possible implications of the work: (i) They suggest their model has the potential to better quantify erosion and infill, and (ii) They confirm the origin of Chryse Planitia as an early impact basin.

After rereading the paper and considering its implications, I am afraid I am still concerned that these results do not merit publication in Nature Communications. (If anything, I think the work is even less appropriate here than for Nat Geo because this journal addresses a broader audience for whom novelty and broader implications should be even more important.)

The paper argues that recognizing Chryse as an impact basin could cause radical changes in our interpretations of events on early Mars could be used to constrain mantle rheology and thermal evolution based on the relaxation state. But such profound conclusions are not presented here. What the paper does offer is a very nice analysis which strengthens (though not totally proves) a previously suggested interpretation. I am sorry I therefore I cannot recommend publication in Nature Communications. I recommend the authors send their work to one of the main stream, high quality journals in Planetary Science, and I do believe it should be published in such a journal.

> We agree with the reviewer that there have been previous suggestions of Chryse originated as an impact basin and we did acknowledge this fact in our manuscript. In this manuscript, we applied a novel and original technique to test this assumption based on high quality datasets and suggest new evidence to support this idea.

The deeper implications for the early evolution of Mars are now further discussed in the final discussion section with comparison to the moon where it is observed that a change in relaxation state occurred early between 4.2-4.4 Gyr. Similar changes may have occurred on Mars and may help constrain the thermal evolution history of the planet. Indeed, these important implications would require significant and thorough analysis of all impact basins on Mars, many of which are not well recognized yet. There is surely more work to be done in that respect, which proves that our manuscript should be of wide interest.

In summary, we believe that our manuscript is appropriate to Nature Communications because it uses a novel technique to provide an answer to an old question which, to us, represents an important advance for specialists in our field and is also of interest to a more general audience. Furthermore, basin relaxation depends on crustal thickness and thermal state that should be better constrained soon thanks to the ongoing InSight mission. Our study is thus a timely and relevant one regarding the interpretation of InSight data.

Reviewer #2 (Remarks to the Author):

The reviewer is highly pleased with the extensive MS reworking by authors. Now most of reviewer's concerns are answered. The main idea of the MS – the possibility of the impact origin of the Chryse depression and fast filling of the original depression with sediments – is now presented in better style than in the first MS' variant. However, a couple of minor problems of the paper perception still exist. In reviewer's opinion, the submitted version 2.0 is good enough for publication, but not meet yet standards of the Nature Communication. The recommendation is a minor editing without the next review (see the wishlist below).

>The reviewer's thoughtful comments are much appreciated and improved the manuscript in the presentation of the problem, the method, and discussion. We here address the remaining concerns of the reviewer with a response to each point. Please see the point-by-point response below.

1. I would advise to change the word "density" to "probability density" through the whole text (both main and supplementary). A general reader in planetology commonly uses "density" or as a physical density, or an "areal density" (number of craters per unit area). Authors frequently have in mind the statistical probability density (the fraction of craters with a given d/D in the narrow interval of d/D).

>Yes, we agree with the reviewer that this should be clarified in the text. This term has now been corrected throughout the article and the terms “probability density” and “areal density” are used to under different suitable contexts, as the reviewer pointed out.

2. The usage of nonparametric probability density estimation has the sense when we have underlying ideas what is the reason for the data scatter. It may be a statistical error of measurements, the difference in the crater degradation rate in various places (sedimentation could be faster in local depressions), or whatever else. It would be great if in Supplements authors extend the explanation of their statistical technique choice. Just now it looks like they have a standard (MATLAB?) computer program and use it because they have it. In the previous review I pointed out the presence of $d/D < 0$ in the statistical processing. Now authors simply put all virtual numbers of craters with $d/D < 0$ into the first (?) d/D bin next to $d/D = 0$. Why authors name this “improvement” as a “reflection technique”? Is it possible to show for the comparison a simple statistics of d/D (for example in a style used for lunar craters by Basilevsky, A. T., Kreslavsky, M. A., Karachevtseva, I. P., and Guskova, E. N., Morphometry of small impact craters in the Lunokhod-1 and Lunokhod-2 study areas, Planet. Space Sci., 2014. V. 92. P. 77-87)? It is non-mandatory, sure, but a short paragraph in Supplementary would greatly improve the paper perception.

>Yes, we agree that this probability approach is based on the thoughts of the underlying uncertainties of measurements and geological processes which may result in some degrees of scattering in the dataset. This was not very clear in our earlier version where we did not detail the rationale behind this, so we improve the introduction to density estimation with more details in the Method section. Given the extended Method section, we detail our rationale for using kernel density estimation as well as the choice of bandwidth and kernel in the section-Methods, Kernel density estimation. We also have written a clearer explanation of the added the “reflection method” which accounts for the negative probability density near $x=0$. This is a more realistic representation of the probability density since depth does not go to negative in reality, and therefore makes an improvement to the probability density estimation of the crater population distribution. We hope these improvements will help with the readers’ perception of the method.